# Physical Inactivity and Sedentary Behaviour among Panamanian Adults: Results from the National Health Survey of Panama (ENSPA) 2019

**DOI:** 10.3390/ijerph20085554

**Published:** 2023-04-18

**Authors:** Roger Montenegro Mendoza, Reina Roa, Flavia Fontes, Ilais Moreno Velásquez, Hedley Quintana

**Affiliations:** 1Department of Research and Health Technology Assessment, Gorgas Memorial Institute for Health Studies, Panama City 0816-02593, Panama; 2Planning Directorate, Ministry of Health, Panama City 4444, Panama; 3Dietetic and Nutrition Department, Faculty of Medicine, University of Panama, Panama City 3366, Panama

**Keywords:** physical inactivity, GPAQ, sedentary behaviour, domain-specific physical activity, Panama

## Abstract

Physical inactivity (PI) has been described as an independent risk factor for a large number of major non-communicable diseases and is associated with an increased risk of premature death. Additionally, sedentary behaviour has been associated with increased overall mortality. We estimated the national prevalence of PI and sedentary behaviour using the Global Physical Activity Questionnaire version 2. Using unconditional logistic regressions, the possible risk factors for PI were assessed. Over half of the people included in this study (54.9%; 95% CI: 54.1–57.3%) were physically inactive, with the median time spent engaged in sedentary behaviour being 120 min per day. Statistically significant associations with PI were observed with regard to sex, living area, and alcohol consumption. PI prevalence in Panama was elevated and showed a sex difference (women: 64.7%, 95% CI: 63.7–66.7%; men: 43.4%, 95% CI: 41.5–47.5%). According to our analysis of three-domain-related physical activities, the main contribution to the total estimated energy expenditure of physical activity/week came from the transport domain, followed by the work/household domain, and the least significant contributor was consistently the domain of exercise- and sports-related physical activities.

## 1. Introduction

Physical activity (PA) is defined as any bodily movement produced by the contraction of skeletal muscle that results in energy expenditure [1]. Regular PA has been associated with a reduced risk of all-cause mortality and several chronic conditions, as well as psychological health [2,3,4]. World Health Organization (WHO) guidelines recommend an average weekly volume of PA of 150 min of moderate intensity, 75 min of vigorous intensity, or an equivalent combination of moderate to vigorous intensity (MVPA) exercise [5]. In contrast, physical inactivity (PI) is defined as a PA level that is insufficient to meet the WHO guidelines’ recommendations [6] and is considered to be an independent risk factor for a large number of major non-communicable diseases (NCDs) [7,8,9]. Sedentary behaviour (SB), on the other hand, is defined by the Sedentary Behaviour Research Network as *“any waking behaviour characterized by an energy expenditure ≤ 1.5 Metabolic Equivalents (METs) while in a sitting or reclining posture”* [10]. A MET is a unit used to estimate the energy expenditure of PA, and it is presented as multiples of the resting metabolic rate [11].

The evaluation of PA includes the assessment of different dimensions, such as the frequency (how often), duration (how long), and intensity (the rate of energy expenditure demanded in METs) performed in different domains such as transport, paid or unpaid work, or exercise and sports activities [11,12,13]. PA can be assessed using objective techniques (i.e., a heart rate monitor, pedometer, or accelerometer) or subjective techniques, such as self-report methods (PA diaries and logs or recall surveys). Nevertheless, for the surveillance of PA in population groups, self-report questionnaires are frequently used as they are less expensive and easier to administer than objective techniques [14].

Aiming to assess both health and disease status, as well as the risk factors associated with the health-related outcomes of the Panamanian population, the National Health Survey of Panama (in Spanish, Encuesta Nacional de Salud de Panamá—ENSPA) was carried out in 2019. In the ENSPA, the Global Physical Activity Questionnaire (GPAQ) version 2 was used to measure PA, PI, and SB [14] at the national level. To the best of our knowledge, before the ENSPA study, in the Republic of Panama, PI was estimated to occur in 7.4% of the population from the provinces of Panama and Colon, where 60.4% of all Panamanians aged 18 years or older resided in 2010 [15]; however, in this survey, PI was measured using only three questions. Thus, very little is known about recent estimates of PI in the Panamanian adult population without knowledge based on a comparable international questionnaire. The characterisation of PI is essential for tracking our progress toward the global target of a relative reduction by 2030 and many of the Sustainable Development Goals (SDG) [16]. Thus, the aims of this study were: (1) to estimate the prevalence of PI; (2) to assess the associations of sociodemographic, health, and lifestyle factors with PI; and (3) to assess SB among the study participants.

## 2. Materials and Methods

### 2.1. Settings

The ENSPA study has been described previously [17,18,19,20]. Briefly, between June and December 2019, participants who had been living in their households for at least six months were invited to participate in the study, which comprised a long questionnaire, anthropometric and blood pressure measurements, and blood sampling. A complex sampling design was applied to select the households and, subsequently, the participants, attempting to achieve representativeness of the results relative to the population of the whole country, including urban, rural, and indigenous areas. Figure 1 shows the inclusion criteria for our study population. In the present study, the sample number to be analysed was 13,982 participants [11,21].

### 2.2. Sociodemographic Characteristics and Variables

Age was measured in years, and the living area was classified as urban, rural, or indigenous. According to the highest education level achieved, individuals were categorised as follows: no formal schooling, special or primary schooling (0–6 years), secondary schooling, short-cycle tertiary or other schooling (7–12 years), and university (≥13 years).

Ethnic group status was self-reported and categorised as Caucasian, Afro-Panamanian, multiracial (mulato, trigueño, culizo, and mestizo), Indigenous, or Asian and others.

### 2.3. Health and Lifestyle Factors

Self-perception of health status was reported by the participants as very good, good, very bad, or bad. Then, for the analysis, we dichotomised the variable as very good or good and very bad or bad.

Ever consumption of tobacco products was assessed using the following questions: “in the past, have you smoked tobacco on a daily basis, less than daily or not at all?” and “in the past, have you used smokeless tobacco on a daily basis, less than daily or not at all?” [22]. Alcohol consumption was assessed through the question: “have you ever consumed alcoholic beverages, including craft beverages, in your life?”

### 2.4. Anthropometric Measurements

Weight and height were measured by two trained health personnel, as described in detail in a previous paper [18]. Body mass index (BMI) was calculated as the person’s weight in kg divided by the square of his/her height in meters. BMI was then categorised as underweight (BMI below 18.5 kg/m^2^), normal weight (BMI between 18.5 kg/m^2^ and <24.9 kg/m^2^), overweight (BMI between 25 kg/m^2^ and <29.9 kg/m^2^), or obese (BMI ≥ 30 kg/m^2^) [23]. Pregnant women without oedema (totalling two hundred and eighty-six) were assessed according to the weight gain pattern of the Atalah curve [24].

### 2.5. Physical Inactivity (PI)

PI was assessed using the GPAQ version 2 in the Spanish language [25]. The GPAQ comprises 16 questions that are used to evaluate the number of days per week (frequency), the total minutes (duration), and the intensity (moderate or vigorous) of an activity for each PA domain (work-/household-, transport-, and exercise- and sports-related physical activities).

For each domain, the estimated energy expenditure (EEE) of PA/week was calculated as the product of the frequency (in number of days) by the duration (in minutes) and the intensity (4 METs were assigned to moderate activities and transport activities, while 8 METs were assigned to vigorous activities in the other two domains). The total EEE of PA/week was defined as the sum of the EEE of PA in all the domains. Individuals who did not perform at least 10 min of either moderate or vigorous activities continuously for each domain demonstrated physically inactive [14].

The total EEE of PA/week was analysed as a continuous and ordinal variable according to the WHO recommendations for adults as follows: high PA (total PA ≥ 3000 MET-minutes/week or vigorous PA ≥ 1500 MET-minutes/week), moderate PA (total PA ≥ 600 MET-minutes/week), or low PA or physically inactive (total PA < 600 MET-minutes/week or no PA in all domains, respectively) [11,26,27,28,29].

### 2.6. Sedentary Behaviour (SB)

SB was measured through a single-item question about the usual amount of time in minutes spent engaged in sitting or reclining activities (e.g., at work, at a desk, at home, travelling or getting to and from places by car or bus, reading, sitting with friends, or watching television) on a typical day [30].

### 2.7. Statistical Analysis

All statistical analyses were performed after applying the inclusion criteria for our study population [11]. The contribution of the domain-specific PA (MVPA at work/the household, exercise and sports activities, and transport) to the total EEE of PA/week was determined per each individual, and then the mean of these percentages was calculated as described in the GPAQ analysis guide [11] and a previous paper [31]. The relative contribution of the domain-specific PA was analysed by sex and sociodemographic variables. For continuous variables, the median and the interquartile range (IQR) were reported. For categorical variables, the prevalence was estimated with their respective 95% confidence intervals (95% CI). For both continuous and categorical variables, the complex sampling design was applied.

Unconditional logistic regression models were calculated to estimate the associations of PI with sex, age, living area, educational level, BMI, self-perception of health status, ever tobacco consumption, and alcohol consumption based on crude models adjusted for all the other regressors mentioned above. The odds ratio (OR) and 95% CI of the model was estimated.

SB was analysed according to the median and IQR considering sex and all sociodemographic and lifestyle variables. To estimate the prevalence of sedentary lifestyles, we also created three dichotomous variables using thresholds reported in previous studies, encompassing ≥ 4 h [32], >7 h [33], and >8 h [34,35] of sedentary activities per day.

The rates of prevalence and general characteristics are presented weighted. All analyses were sex-stratified, and the general characteristics of the excluded individuals with missing data are described in the Appendix A. The calculations were performed using STATA software (version 14; Stata Inc., College Station, TX, USA).

## 3. Results

### 3.1. Characteristics of Participants

Figure 1 depicts a flowchart summarising the exclusion and inclusion criteria for this study. After applying the exclusion criteria, the expanded sample for the present study encompassed 2,192,075 participants nationwide. Those excluded did not differ much with respect to their baseline characteristics from those included in the analysis, as presented in the Appendix A.

Table 1 presents the baseline characteristics of the study participants by sex. Women were younger (median 39 years old, IQR: 28–53) than men (median: 43 years old, IQR: 30–56), with a statistically significant difference (*p*-value: <0.05). Education level differed between the sexes; however, more than a half of all the participants reported having a secondary education, and a higher proportion of women reported having a university education. The proportion of obesity and rate of very bad or bad self-perception of health status were higher among women than men, while ever tobacco and alcohol consumption was reported more frequently by men.

In Figure 2, the weighted standardised prevalence of the low physical activity or physically inactive category and physical activity category stratified by sex is shown. Just over half of the individuals were physically inactive, with the highest frequency being among women (64.7%; 95% CI: 63.7–66.7%) rather than men (43.4%; 95% CI: 41.5–47.5%) (*p*-value < 0.001). In contrast, men had a 2.8 times higher prevalence of high PA (39.3%; 95% CI: 35.5–41.0%) than women (14.1%; 95% CI: 12.8–15.0%) (*p*-value < 0.001). We also describe the weighted standardised prevalence of the physical inactivity and physical activity categories stratified by age in the Appendix A.

Table 2 displays the prevalence of study participants with no PA by domain and in any domain and with no vigorous-intensity PA, according to sex. Overall, the exercise and sports domain presented the highest prevalence of no PA, followed by the work/household domain. Even though this pattern was observed in both sexes, for all the domains and for no vigorous-intensity PA, women had a higher prevalence of no PA than men (*p*-value < 0.01). We also describe the prevalence of no physical activity by domain and in any domain and no vigorous-intensity physical activity stratified by age in the Appendix A.

Table 3 presents the median and IQR of PA measured in MET-minutes/week by domain and in total, according to sex. The highest EEE median was reported for the work/household domain, followed by the exercise and sports and transport domains. For each domain, women presented a lower median EEE of PA/week than men. The overall EEE medians of PA/week and the work/household domain were 3.9 and 4.7 times higher in men than in women, respectively. For the exercise and sports domain, the EEE median was 2.0 times higher in men than in women, while for the transport domain, the EEE median was 1.6 times higher in men than in women.

Table 4 presents the mean relative contribution of each PA domain to the total EEE of PA/week among participants who engaged in at least 10 min of continuous PA, according to sex and sociodemographic characteristics. For all the analysed sociodemographic characteristics and for both sexes, the main contribution to the total estimated EEE of PA/week was that of the transport domain, which increased with age and habitation in urban areas compared to indigenous areas. Additionally, among men, the transport domain’s contribution to the total EEE of PA/week increased with the educational level achieved, except among women where it was decreased.

The work/household domain was the second largest contributor to the total EEE of PA/week, being similar among men living in rural and indigenous areas but higher than those living in urban areas. Among women, the highest contribution of the work/household domain was observed for those living in indigenous areas, with statistically significant differences observed when compared with their peers living in the other two types of areas. Concerning the educational level among men, the contribution of the work/household domain to the total PA decreased with respect to the educational level achieved, with statistically significant differences between the various educational categories. Among women, the same trend was observed, but a statistically significant difference was only observed for the lowest educational level.

In both sexes, the exercise and sports domain’s contribution to the total EEE of PA/week decreased with age, increased with the education level achieved, and was higher in urban areas. Overall, in the case of both sexes, those participants classified as having moderate and high PA showed that the largest contributions were those of the work/household and exercise and sports domains, while for those classified as having low PA, the largest contribution to the total PA/week was that of the transport domain.

### 3.2. Regression Analysis

Table 5 presents the crude and adjusted associations between the individual characteristics and physical inactivity. Women had a higher probability of being classified as inactive in comparison to men, and individuals living in rural and indigenous areas had a lower probability of being classified as physically inactive in comparison with people living in urban areas. Lastly, individuals reporting ever alcohol consumption had a lower probability of being physically inactive than persons who declared that they never drink alcoholic beverages. Although we could not demonstrate a statistically significant association between the education level achieved and PI, the confidence intervals for the point estimates of the two first education levels were marginal.

In general, the reported time engaged in SB was 120 min, irrespective of sex and age group. However, individuals living in indigenous areas reported values amounting to half of this measurement in comparison with those living in urban and rural areas. For both sexes, the median value of SB increased with education level. The results are presented in the Appendix A
Appendix A. With respect to the prevalence of SB, our results showed that the sitting time varied, including values such as 27.9% (95% CI 26.5–29.3) (≥4:00 h), 8.4% (95% CI 7.5–9.4) (>7:00 h), and 3.4% (95% CI 2.8–4.2) (>8:00 h). The results are presented in the Appendix A
Appendix A.

## 4. Discussion

### 4.1. Main Findings

This study explored nationwide features of PI among individuals aged 18–69 years in Panama. Overall, more than half of the study population did not meet the WHO recommendations for PA; hence, they were found to be at a health risk due to PI. According to the regression analysis, the sociodemographic and lifestyle factors associated with PI included being female, while people living in rural and indigenous areas, as well as those who indicated ever alcohol consumption were less likely to be classified as physically inactive. In addition, among the participants of this study, the estimated median of daily sedentary time was low in indigenous areas, while the highest median was observed for those with a university-level education.

The PI prevalence reported in this study was higher than the worldwide estimation (27.5%), as well as the figure for the Latin American and Caribbean (LAC) region (39.1%) [36]. However, it was similar to the mean prevalence in India between 2008 and 2010 for individuals aged ≥20 years (54.4%) [37] and the two countries with the highest prevalence worldwide (American Samoa at 53.4% and Kuwait at 67.0%) [36]. When comparing our results with those of other countries using the same questionnaire and methodology for an analysis of adult populations, the prevalence of PI in Panama was still higher than estimations for Iran in 2011 (44.8%), Chiang Mai, a province of Thailand in 2014 (26.0%), Armenia in 2016 (21.6%), Singapore in 2019–2020 (16.7%), and Nepal in 2013 (3.0%) [35,38,39,40].

In agreement with previous studies, we observed a sex difference in PI [36,37,41,42]. Furthermore, among women who met the WHO weekly recommendation for PA, only a small proportion performed a high level of PA. Although gross differences were observed in our estimates of non-PA between men and women, in general, both reported engaging mostly in transport-related PA, with the lowest proportion represented by exercise- and sports-related PA. However, most of the EEE of PA/week was related to the work/household domain, while the transport domain was the least, which was somewhat expected since transport-related activities are always considered moderate in intensity.

When analysing relative domain-specific contributions to the overall EEE of PA/week, independent of sex and age, the transport domain was the highest and the exercise and sports domain was the lowest of the three (Transport > Work/household > Exercise and sports). However, in a meta-analysis assessing domain-specific contributions to the overall EEE of PA/week in 104 countries, the global ranking order was Work/household > Transport > Exercise and sports, with differences based on sex and age. Among the countries included in the meta-analysis, one of five had the same ranking order as that reported in the current study, and half were classified by the World Bank as upper-middle-income countries in 2020, an income category that Panama also belongs to [42]. Although exercise and sports physical activities are important for cardiovascular and mental health, our results show that these activities were the least frequently performed, and they had the lowest relative contribution to the overall EEE of PA/week in this study. These findings are consistent with the results for 91 of the 104 countries analysed in the aforementioned meta-analysis [42]. In connection with these findings, further analyses should be carried out to identify the determinants of PA performed in this domain.

Our results show that ever alcohol drinking decreases the prevalence of PI, and this finding is in agreement with other reports measuring this association [43,44]. However, the association between alcohol use and PI can be explained by reverse causality, since some drinkers might consider countering the damage induced by alcoholic beverages by performing PA. Another possible explanation is that the alcohol industry is allowed to sponsor exercise and sport activities, which, in turn, might foster performance of those activities and thus increase alcohol consumption among those doing exercise and sports.

The fact that the lowest prevalence of PI was observed in both rural and indigenous areas might be related to the fact that jobs requiring more manual labour are frequently found in those areas. In 2016, the National Institute of Statistics and Census of Panama reported that in the rural areas, 63.9% of people aged over 15 years or older were employed in the primary and secondary sectors of the economy [45]. These results are consistent and supported by the fact that the relative contribution of the work/household domain to the total PA was highest among individuals living in both rural and indigenous areas. Although a previous analysis conducted in the Chilean population found a higher probability of being classified as having PI among those with low education levels [46], in this population, we could not demonstrate a statistically significant association between these two variables; however, given the marginal confidence intervals observed, this could change with a larger study sample.

Although a high prevalence of PI was observed in the Panamanian population, the prevalence of SB, using 7-h thresholds, was similar to the value measured in the Mexican population in 2018 (11.3% vs. 10.0%) [33] and lower than that reported in a Chilean study in 2009–2010 (28.3% vs. 46.6%) where a 4-h threshold was used [32]. When using the 8-h thresholds, the prevalence of sitting time remained lower in Panama (8.5%) than in the Armenian adult population in 2016 (13.2%) [35] and US population in 2015–2016 (25.7%) [34]. In the case of our results, the median value of SB was 2.5 times lower than the value from an analysis of European adults in 2013 [47], and the highest median of SB was observed among individuals with a university education, who are likely to perform office jobs and occupy more managerial positions relative to the third sector of the economy [48]. In general, the SB results were not consistent with the estimated prevalence of physical inactivity, especially when we consider the fact that even individuals classified as physically active could also meet the sedentary criterion. This inconsistency should be addressed by evaluating SB in a more comprehensive manner, for example, by including questions related to the average time spent performing activities such as watching television, working in front of a screen, and even commuting via modes other than walking.

### 4.2. Limitations of the Study

The cross-sectional nature of this study limited our ability to evaluate causality between sociodemographic characteristics and PI. The self-reported data relied on the participants’ memory; thus, their answers could have been subject to recall bias. Moreover, the participants’ understanding of the concept of physical intensity (moderate and vigorous), the social desirability of certain responses, and finally, the subjective technique used to evaluate PA (GPAQ) could have led to overestimations of activity [49] and, thus, misclassification bias. Nevertheless, the interviewers were trained on the use of the questionnaire and read examples of different intensities of physical activity for each domain. Notably, implausible values were excluded from the analysis, as recommended by the GPAQ analysis guidelines. In previous studies, the SB assessment was evaluated through the same single-item question. Nevertheless, due to the low criterion validity (r_w_ = 0.23) reported in a previous meta-analysis, it is recommended to use multi-item SB questionnaires and smart trackers rather than single-item questions [50]. However, international population surveys often rely on this methodology due to their low cost, ease of use, and consequent ability to be implemented on a large scale.

Furthermore, it should be considered that the GPAQ version 2 does not include light-intensity activities and is based on a period of at least 10 min in length for the evaluation of each of the three domains. In contrast, this minimum time period is no longer included in the WHO 2020 guidelines on physical activity and sedentary behaviour, in which the message “Everything counts” is now underscored [6]. As a consequence, the incorporation of shorter bouts of activity and light-intensity activities might affect the relative domain-specific contributions, as well as the total physical activity estimation [51]. Finally, although the GPAQ has been validated extensively in various populations, it has not specifically been validated for the Panamanian population.

### 4.3. Strength of the Study

This analysis is based on a national representative sample using a standardised PA questionnaire. Even though the validation of the GPAQ shows poor to moderate agreement with the accelerometer for MVPA, it is considered a suitable and acceptable instrument for monitoring PA in population health surveillance systems and is currently used by over 100 countries [49,52]. Moreover, the GPAQ provides reproducible data and shows a moderate to strong positive correlation with the international physical activity questionnaire (IPAQ), a previously validated and accepted measure of PA [53,54].

The GPAQ assesses PA in three domains (work/household, transport, and exercise and sports). This characteristic allows for a more accurate analysis of PA in the population rather than concentrating solely on the exercise and sports domain [52]. Furthermore, the analysis of the relative contributions of domain-specific activities to the overall EEE of PA/week allows for a more detailed understanding of PA patterns in the population.

## 5. Conclusions

Our results show that the PI prevalence in Panama was higher than the estimation for the LAC region. The sociodemographic factors that showed statistically significant associations with PI were female sex, living in urban areas, and ever alcohol consumption, while BMI, self-perception of health status, and ever tobacco consumption did not present statistically significant associations with PI. Among each of the three domain-specific PA contributions analysed, exercise- and sports-related PA was consistently the least significant contributor to the total EEE of PA/week. Future research aiming to identify possible determinants of, or barriers to, the practice of PA in each domain, as well as PA among the elderly group in our population, is warranted to enable efficient development of national policies and health promotion methods.

This study provides baseline information on PI and sedentary behaviour based on the GPAQ, which has been implemented on the nationwide level in the adult population in Panama. Further research is necessary in order to continue monitoring PI and SB trends and progress the global action plan aiming to meet the physical activity target for 2030 in the Panamanian population.

## Figures and Tables

**Figure 1 ijerph-20-05554-f001:**
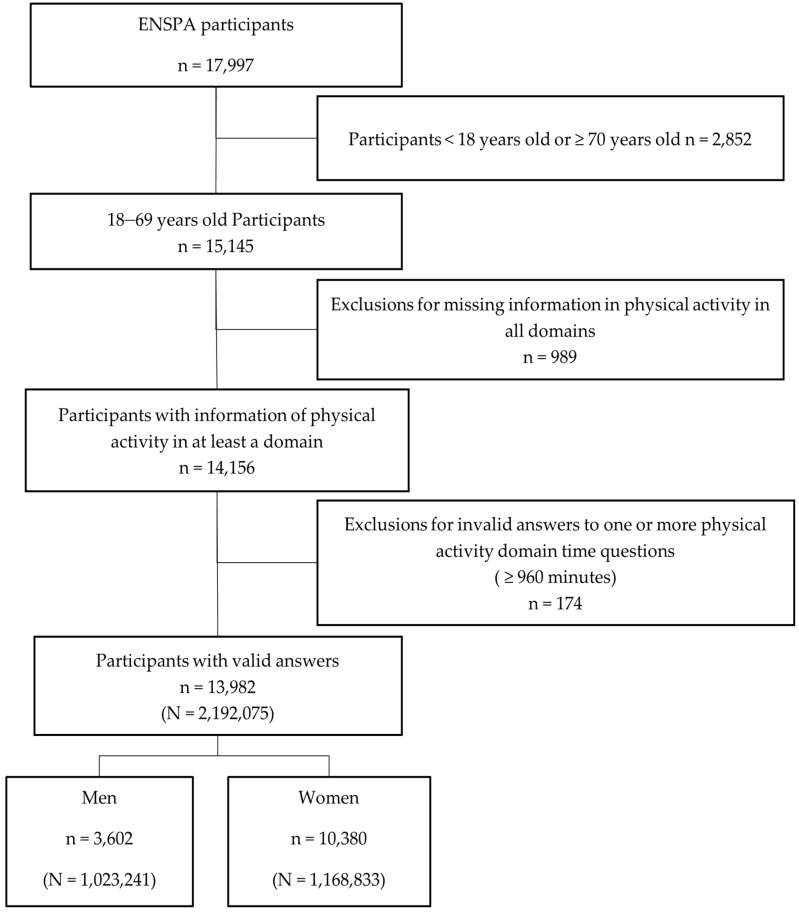
Flowchart summarising the exclusion and inclusion criteria for the study population and weighted frequencies of the study participants.

**Figure 2 ijerph-20-05554-f002:**
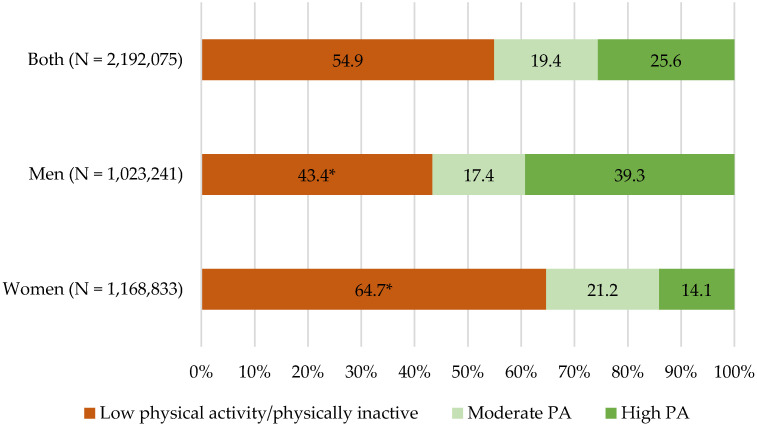
Weighted standardised prevalence of the low physical activity category/physically inactive and physical activity (PA) categories by sex among participants aged 18–69 years. Panama, 2019. * *p*-value < 0.001.

**Table 1 ijerph-20-05554-t001:** Selected baseline characteristics of the study participants aged 18-69 years by sex. Panama, 2019.

Sociodemographic Characteristics	All		Men		Women		*p*-Value
	N	Weighted Prevalence % (95% CI)	N	Weighted Prevalence % (95% CI)	N	Weighted Prevalence % (95% CI)	
	2,192,075		1,023,241	46.7 (45.1–48.3)	1,168,833	53.3 (51.7–54.9)	
Age (years)							<0.001
18–29	579,891	26.5 (25.1–27.9)	244,670	23.9 (21.5–26.5)	335,221	28.7 (27.3–30.1)	
30–39	447,146	20.4 (19.2–21.6)	188,237	18.4 (16.4–20.5)	258,909	22.1 (20.9–23.5)	
40–49	437,647	19.9 (18.8–21.2)	215,550	21.1 (19.0–23.3)	222,097	19.0 (17.8–20.3)	
50–59	385,816	17.6 (16.4–18.8)	192,600	18.8 (16.8–21.1)	193,216	16.5 (15.4–17.8)	
60–69	341,575	15.6 (14.5–16.7)	182,185	17.8 (15.9–19.9)	159,391	13.6 (12.6–14.8)	
Living area							
Urban	1,400,736	63.9 (62.8–64.9)	647,406	63.3 (61.4–65.1)	753,330	64.4 (63.4–65.5)	0.564
Rural	661,303	30.2 (29.2–31.2)	314,670	30.7 (29.0–32.5)	346,633	29.7 (28.7–30.6)	
Indigenous	130,035	5.9 (5.6–6.3)	61,165	6.0 (5.5–6.5)	68,871	5.9 (5.6–6.2)	
Highest education level achieved							
No formal schooling/special or primary schooling (0–6 years)	551,766	25.3 (24.1–26.4)	257,109	25.2 (23.3–27.3)	294,657	25.3 (24.0–26.5)	0.009
Secondary/short-cycle or other schooling (7–13 years)	1,235,178	56.5 (55.1–58.0)	595,419	58.5 (55.8–61.1)	639,759	54.9 (53.3–56.4)	
University (≥13 years)	397,111	18.2 (17.0–19.4)	165,607	16.3 (14.3–18.5)	231,504	19.9 (18.6–21.2)	
Ethnic group							
Afro-Panamanian	336,101	15.3 (14.2–16.5)	173,097	16.9 (14.9–19.1)	163,004	14.0 (12.9–15.1)	0.037
Multi-racial	1,133,402	51.7 (50.2–53.3)	528,794	51.7 (49.0–54.5)	604,608	51.8 (50.2–53.3)	
Indigenous	257,752	11.8 (11.0–12.5)	110,548	10.8 (9.7–12.1)	147,204	12.6 (11.7–13.5)	
Caucasian	399,039	18.2 (17.0–19.5)	183,821	18.0 (15.9–20.3)	215,218	18.4 (17.1–19.7)	
Asian and others	64,417	2.9 (2.5–3.5)	26,187	2.6 (1.8–3.5)	38,231	3.3 (2.7–4.0)	
BMI *							
Underweight	44,309	2.3 (1.8–2.9)	22,679	2.6 (1.7–3.9)	21,630	2.1 (1.6–2.6)	<0.001
Normal	473,461	24.4 (23.0–25.8)	251,183	28.4 (25.9–30.9)	222,278	21.1 (19.8–22.4)	
Overweight	688,819	35.5 (33.9–37.1)	339,893	38.4 (35.5–41.4)	348,926	33.1 (31.5–34.6)	
Obesity	734,249	37.8 (36.3–39.4)	271,875	30.7 (27.9–33.6)	462,375	43.8 (42.1–45.5)	
Self-perception of health status							
Very good or good	1,943,147	88.7 (87.7–89.6)	929,379	90.9 (89.2–92.3)	1,013,768	86.8 (85.7–87.8)	<0.001
Very bad or bad	247,564	11.3 (10.4–12.3)	93,067	9.1 (7.6–10.8)	154,497	13.2 (12.2–14.3)	
Ever tobacco consumption							
Yes	195,012	8.9 (8.0–9.9)	159,985	15.6 (13.9–17.6)	35,028	3.0 (2.5–3.6)	<0.001
No	1,994,862	91.1 (90.1–91.9)	862,462	84.3 (82.4–86.1)	1,132,400	97.0 (96.4–97.5	
Ever alcohol consumption							
Yes	749,008	34.2 (32.7–35.6)	484,229	47.3 (44.6–50.1)	264,779	22.6 (21.4–24.0)	<0.001
No	1,443,067	65.8 (64.3–67.3)	539,013	52.7 (49.9–55.4)	904,054	77.3 (76.0–78.6)	

Source: National Health Survey of Panama (ENSPA) 2019. * BMI: body mass index. Fifty-three (53) participants had missing values for educational level. Seven (7) participants presented with missing values for self-perception of health status and the ethnic group question. In the BMI analysis, a total of one thousand five hundred and seventy (1570) participants were excluded (one thousand two hundred and forty-four (1244) had missing values for weight or height, and three hundred and twenty-six (326) presented with oedema). Twelve (12) participants had missing data for the tobacco consumption question. Multi-racial includes: mulato, trigueño, culizo, and mestizo. Percentages are based on weighted data.

**Table 2 ijerph-20-05554-t002:** No physical activity by domain and in any domain and no vigorous-intensity physical activity stratified by sex among participants aged 18–69 years. Panama, 2019.

Domain	Both N = 2,192,075	Men N = 1,023,241	Women N = 1,680,833	*p*-Value
	Weighted Prevalence % (95% CI)	Weighted Prevalence % (95% CI)	Weighted Prevalence % (95% CI)	
Transport	48.7 (47.2–50.2)	46.1 (43.3–48.9)	51.0 (49.4–52.6)	0.003
Work/household	80.0 (78.8–81.2)	71.3 (68.9–73.6)	87.6 (86.6–88.6)	<0.001
Exercise and sports	82.6 (81.4–83.8)	75.7 (73.4–77.8)	88.6 (87.5–89.6)	<0.001
In any domain	40.2 (38.7–41.8)	34.4 (31.6–37.2)	45.4 (43.8–47.0)	<0.001
Intensity				
No vigorous-intensity PA	77.4 (76.1–78.7)	64.2 (61.6–66.7)	89.1 (88.1–90.0)	<0.001

Source: National Health Survey of Panama (ENSPA) 2019. CI: Confidence interval.

**Table 3 ijerph-20-05554-t003:** Median and interquartile range (IQR) of the estimated energy expenditure (EEE) of physical activity (PA) measured in MET-minutes/week by domain and in total according to sex among participants aged 18–64 years. Panama, 2019.

Domain	EEE of PA (MET-Minutes/Week)						
	Both		Men		Women		*p*-Value
	N	Median (IQR)	N	Median (IQR)	N	Median (IQR)	
Work/household	438,079	3840 (960–11,520)	293,487	6720 (1920–14,400)	144,591	1440 (480–4320)	<0.001
Exercise and sports	381,277	1920 (720–4320)	248,437	2400 (960–5040)	132,840	1200 (480–2880)	<0.001
Transport	1,124,461	840 (360–1920)	551,485	960 (420–3360)	572,975	600 (280–1680)	<0.001
Overall	1,309,991	1800 (600–6720)	671,635	3720 (1120–11,060)	638,356	960 (420–2880)	<0.001
Intensity							
Vigorous-intensity PA	494,214	3360 (1440–9600)	366,366	4800 (1440–11,520)	127,848	1920 (720–4320)	<0.001

Source: National Health Survey of Panama (ENSPA) 2019. IQR: interquartile range; EEE: estimated energy expenditure; PA: physical activity; MET: metabolic equivalent task. Respondents who declared at least 10 min of continuous PA/week in at least one domain.

**Table 4 ijerph-20-05554-t004:** Relative domain contribution to the total estimated energy expenditure (EEE) of physical activity (PA)/week among individuals aged 18–69 years engaging in at least 10 min of continuous PA, according to sex and the selected variables. Panama, 2019.

Variables	Sex							
	Men % (95% CI)				Women % (95% CI)			
	N	Work/Household	Exercise and Sports	Transport	N	Work/Household	Exercise and Sports	Transport
	671,635	31.4 (29.9–33.0)	21.6 (20.3–22.9)	47.0 (45.3–48.6)	638,356	14.8 (14.0–15.5)	12.9 (12.1–13.6)	72.3 (71.4–73.3)
Age (years)								
18–29	170,065	26.1 (23.3–29.0)	36.0 (33.0–38.9)	37.9 (34.9–40.8)	183,373	13.7 (12.4–15.1)	15.4 (13.9–16.8)	70.9 (69.1–72.7)
30–39	120,092	28.8 (25.4–32.2)	25.3 (22.2–28.5)	45.8 (42.2–49.5)	149,169	17.4 (15.8–19.0)	12.9 (11.5–14.3)	69.7 (67.7–71.6)
40–49	155,100	37.9 (34.4–41.5)	15.1 (12.5–17.7)	46.9 (43.3–50.5)	120,033	14.5 (12.8–16.3)	12.9 (11.3–14.5)	72.5 (70.3–74.7)
50–59	112,867	37.9 (33.9–41.9)	15.2 (12.3–18.1)	46.9 (42.9–50.9)	106,258	13.3 (11.4–15.1)	11.0 (9.2–12.9)	75.7 (73.2–78.1)
60–69	113,511	26.9 (23.3–30.4)	11.2 (8.8–13.6)	61.9 (58.0–65.8)	79,523	14.6 (12.2–17.0)	9.3 (7.3–11.3)	76.1 (73.1–79.1)
Living area								
Urban	389,194	24.8 (22.4–27.1)	25.2 (22.9–27.5)	50.0 (47.3–52.6)	384,591	13.4 (12.3–14.6)	14.3 (13.1–15.5)	72.3 (70.8–73.9)
Rural	236,265	40.4 (38.0–42.8)	16.4 (14.7–18.1)	43.2 (40.9–45.5)	213,126	15.3 (14.1–16.4)	11.0 (10.1–12.0)	73.6 (72.3–75.0)
Indigenous	46,176	41.9 (38.0–45.9)	17.3 (14.3–20.3)	40.8 (36.7–44.8)	40,639	25.1 (22.5–27.7)	9.1 (7.4–10.7)	65.9 (62.9–68.8)
Highest education level achieved								
No formal schooling/special or primary schooling	168,922	42.7 (40.0–45.5)	12.0 (10.2–13.7)	45.3 (42.6–48.0)	162,622	16.6 (15.2–18.0)	8.1 (7.1–9.1)	75.3 (73.7–76.9)
Secondary/short-cycle or other schooling	382,214	30.8 (28.6–33.0)	23.1 (21.2–25.1)	46.0 (43.7–48.3)	343,446	13.9 (12.8–15.0)	13.6 (12.6–14.7)	72.4 (71.0–73.8)
University	116,235	16.1 (12.9–19.4)	30.9 (26.9–34.8)	53.0 (48.7–57.3)	130,633	14.1 (12.3–15.8)	16.9 (15.0–18.8)	69.0 (66.6–71.4)
Ethnic group								
Afro-Panamanian	110,905	34.2 (30.1–38.5)	20.6 (17.2–24.0)	45.1 (40.9–49.3)	89,928	12.2 (10.2–14.2)	11.4 (9.5–13.4)	76.3 (73.7–79.0)
Mixed ethnicities	355,021	31.0 (28.8–33.3).	23.1 (21.1–25.1)	45.8 (43.5–48.2)	353,651	14.7 (13.6–15.7)	13.8 (12.8–14.8)	71.5 (70.2–72.9)
Indigenous	79,009	41.5 (38.1–44.8)	14.8 (12.5–17.1)	43.8 (40.4–47.1)	81,282	20.7 (18.8–22.7)	9.6 (8.2–11.0)	69.6 (67.4–71.9)
Caucasian	111,346	22.6 (18.8–26.4)	22.1 (18.3–25.9)	55.2 (50.6–59.9)	99,805	11.9 (10.1–13.8)	14.0 (12.1–16.0)	74.0 (71.5–76.6)
Asian and others	14,559	28.5 (18.7–38.2)	25.9 (16.2–35.7)	45.5 (34.5–56.6)	13,122	16.6 (10.7–22.5)	9.4 (4.7–14.0)	74.0 (66.6–81.4)
BMI								
Underweight	12,892	12.0 (3.8–20.1)	41.6 (28.9–54.3)	46.4 (33.3–59.5)	10,832	13.6 (8.5–18.6)	15.8 (10.3–21.2)	70.7 (63.5–77.8)
Normal	180,503	33.2 (30.4–36.1)	26.2 (23.7–28.7)	40.5 (37.7–43.3)	124,593	14.7 (13.1–16.4)	15.2 (13.5–16.9)	70.1 (67.9–72.3)
Overweight	221,770	36.3 (33.5–39.1)	20.5 (18.2–22.7)	43.2 (40.4–46.0)	194,946	15.5 (14.1–16.9)	12.6 (11.3–13.8)	71.9 (70.2–73.7)
Obesity	172,488	28.0 (24.8–31.2)	17.0 (14.4–19.6)	54.9 (51.4–58.4)	247,081	15.1 (13.9–16.4)	13.1 (11.9–14.3)	71.8 (70.1–73.4)
Self-perception of health status								
Very good or good	616,405	30.9 (29.3–32.5)	22.2 (20.8–23.6)	46.9 (45.2–48.6)	553,125	13.2 (12.4–14.7)	13.0 (12.2–13.8)	73.0 (72.0–74.1)
Very bad or bad	54,436	36.3 (30.6–42.0)	15.1 (11.1–19.1)	48.6 (42.8–54.4)	84,662	19.6 (17.1–22.0)	12.2 (10.1–14.3)	68.3 (65.3–71.3)
Ever tobacco consumption								
Yes	367,220	34.1 (32.1–36.2)	22.5 (20.7–24.3)	43.3 (41.3–45.4)	176,451	15.6 (14.1–17.1)	14.6 (13.2–16.0)	69.8 (67.9–71.6)
No	304,415	28.2 (25.9–30.5)	20.5 (18.4–22.5)	51.3 (48.8–53.9)	461,906	14.5 (13.6–15.3)	12.2 (11.4–13.0)	73.3 (72.2–74.5)
Physical activity								
Low	99,048	4.5 (2.6–6.4)	8.5 (5.7–11.2)	87.0 (83.8–90.3)	225,748	6.2 (5.3–7.2)	4.7 (3.9–5.6)	89.0 (87.8–90.3)
Moderate	178,143	17.7 (15.3–20.2)	18.9 (16.4–21.3)	63.4 (60.4–66.4)	247,441	12.2 (11.1–13.3)	13.8 (12.6–15.0)	73.9 (72.4–75.4)
High	394,444	44.4 (42.3–46.5)	26.1 (24.3–27.9)	29.5 (27.7–31.2)	165,467	30.3 (28.4–32.1)	22.6 (20.9–24.3)	47.2 (45.2–49.1)

Source: National Health Survey of Panama (ENSPA) 2019. CI: confidence intervals. Values are presented as frequencies and values of the weighted prevalence with the respective 95% CI. Physical activity: low (<600 METs-minutes/week); moderate (600–1499 METs-minutes/week); high (≥1500 METs-minutes/week).

**Table 5 ijerph-20-05554-t005:** Adjusted associations between individual characteristics and physical inactivity among participants aged 18–69 years, presented as odds ratios (OR) and 95% confidence intervals (CI). Panama, 2019.

Variables	OR (95% CI)	
	Crude	Adjusted (*n* = 1,932,405)
Sex		
Men	(Ref)	(Ref)
Women	2.33 (2.04–2.65)	2.07 (1.77–2.42)
Age (years)	1.01 (1.00–1.01)	1.00 (0.99–1.01)
Living area		
Urban	(Ref)	(Ref)
Rural	0.67 (0.59–0.76)	0.67 (0.58–0.78)
Indigenous	0.70 (0.59–0.83)	0.63 (0.52–0.77)
Highest education level achieved		
No formal schooling/special or primary schooling	1.08 (0.91–1.29)	1.21 (0.97–1.51)
Secondary/short-cycle or other schooling	1.08 (0.91–1.29)	1.17 (0.97–1.42)
University	(Ref)	(Ref)
Body mass index (BMI)	1.02 (1.01–1.03)	1.01 (0.99–1.02)
Self-perception of health status		
Very good or good	(Ref)	(Ref)
Very bad or bad	1.31 (1.08–1.58)	1.22 (0.99–1.51)
Ever-consumption of tobacco		
No	(Ref)	(Ref)
Yes	0.51 (0.40–0.65)	0.83 (0.63–1.09)
Ever-consumption of alcohol		
No	(Ref)	(Ref)
Yes	0.48 (0.42–0.55)	0.59 (0.50–0.69)

Source: National Health Survey of Panama (ENSPA) 2019.

## Data Availability

The datasets used and/or analysed during the current study are available from the authours on reasonable request via email.

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
