# Peer review of "Physical Inactivity and Sedentary Behaviour among Panamanian Adults: Results from the National Health Survey of Panama (ENSPA) 2019"

_ijerph, 2023, doi:10.3390/ijerph20085554_

Round 1

Reviewer 1 Report

The aim of the study is to define the prevalence of physical inactivity and sedentary behaviour and how several factors might influence this prevalence in the Panamanian population. 

First, I would like to congratulate authors for the study performed. Epidemiological studies involving this type of analysis are necessary to analyse the population behaviour and, therefore, to establish public health policies for the promotion of physical activity. 

Following, I provide some observations and recommendations for improving your manuscript:

General:

Check grammar and punctuation. For example: Line 17: Check spelling: Insert a comma after regressions; or change persons to people.

- Line 18: PI refers to physical inactivity; and in that sentence you mean physically inactive (adjective). Change, please, so that the sentence has the correct spelling.

Abstract:

- Enter abbreviations (e.g. EEE, PA...).

The abstract may not repeat the exact introduction sentences. 

- Order the results: First the PA values, then the percentages and then the subsequent analyses.

Introduction:

The introduction is not synthesised and very fragmented. There are paragraphs that need to be joined together.

Lines: 28-49: The definitions of the 3 behavioural components and their relationship to health should be synthesised and summarised in a single paragraph. The concepts are the journal's own topics, so readers do not need a specific explanation of each of them. 

The definition of physical activity is misquoted (Caspersen, et al., 1985; PMC1424733). 

Lines 50 to 67: the information is adequate but should be combined and synthesised in a single paragraph.

Authors should mention articles already published on the Panamanian population (either their own or from other research groups) and explain what is already known about the Panamanian population and justify the need to know more about the behaviour of the population, and thus support the objective of their paper.

Line 81: verify the verbs properly. 

Methods:

Although the study has been carried out with a larger sample, if for this article only the adult sample has been used, it is recommended not to mention the ages of the overall study in the text. It is recommended to modify lines 90 to 95, only explaining that a data-cleaning of the complete sample was carried out (and reference to figure 1) to select the cases that met the inclusion criteria of this work and that therefore, the sample to be analysed was... 

This avoids repeating information in the text that is already explained in the figure (as requested by the journal).

Regarding the indication of data-cleaning, the reference provided does not describe this, or at least the reviewers have not found it. Could the authors provide references justifying this measure?

Line 119 followed by 118.

Lines 136-140: Overall EEE of PA/week was considered in absolute values and was classificated according to the WHO recommendations for adults: (i) high (Total PA ≥ 3000 MET-minutes/week or a ....

161: To explain more precisely the SB analysis. 

RESULTS:

In general, the quantitative data expressed in the figures are suggested not to be repeated in the text. Just reference the figure/table and provide an overall interpretation or summary of them (without statistics, as these can be consulted in the table/figure).

In the text, several gender comparisons are made (e.g., women are younger than men). Have the authors carried out the relevant analysis to establish these? If so, report the results in the tables.

A Pearson's Chi-square test (χ2) analysis for the variables analysed is recommended and reported in the figures/tables.

For the subsequent tables, why have the authors chosen this type of analysis, differentiating it between the groups?

Again, it is suggested that numerical values (already presented in the tables) are not reported in the text, and interpretations (justified with relevant comparative analysis, if done) are maintained.

Line 209 (do not abbreviate if it will not be used in the table more than once).

Line 257: Sub-heading or appropriate wording.

In table 5, indicate the significant factors in the analysis.

To facilitate the interpretation of the results, it is suggested that the largest tables might be reported in the supplementary material and only the significant results should be included in the text. For smaller tables (e.g., table 3) it is suggested that they be presented in the manuscript (as the authors have done) but for all groups.

DISCUSSION.

The discussion is very fragmented (remember that a paragraph should show a point and that it should be composed of at least three sentences).

Why do the authors mention "estimated prevalence"?

Line 292: Why do the authors compare their results with these countries?

Authors should order the comparative results, starting first with global data, followed by regional data, and then with country-specific data. They should also provide reasons and/or hypotheses based on the literature for these differences.

The authors highlight a gender gap, but has this been statistically tested?

The ideas they subsequently show, as well as the interpretations and contrasts with other articles, are adequate and very relevant. However, they continue to be presented in a fragmented way, which does not facilitate interpretation.

With respect to the analysis of sedentary behaviour, it is unbalanced in comparison to PI, and more discussion is needed for comparisons and interpretation of this variable.

Finally, it is recommended that hypotheses and interpretations are argued with relevant references. For example, line 346-347, "This finding could be explained by the type of occupation that could include more sitting activities such as office work". Authors are correct with this observation, as it is very well stated, therefore, including some reference to previous work that supports this hypothesis is recommended.

CONCLUSIONS:

Line 388: the results show, not suggest.

For gender assertions, again the need to report these statistically significant differences is emphasised.

References: Both the format of citations and references is wrong. Check journal requirements.

There are references linked to websites, which should be referenced to timely scientific articles (e.g., GPAQ reference).

Finally, to complement the analysis it is suggested to perform the following analyses:

- Make a figure like figure 2 with the stratification by age groups.

- Considering the results of the regression analysis, analyse the differences between the significant factors.

- Carry out the same analysis as in table 2, for the rest of the groups and stratified by age.

In conclusion, I would like to congratulate the authors again for their work. It is not easy to compile and synthesise all the information from this type of studies, so I hope that my observations will help you to improve the manuscript.

Reviewer 2 Report

This study estimates the national prevalence of PI and SB in the Republic of Panama. This is a meaningful study, but there are several revisions that should be recommended in the manuscript, as follows.

Title: Prevalence of Physical Inactivity and Sedentary Behaviour in Panamanian Adults Using the Global Physical Activity Questionnaire (GPAQ). Results from the National Health Survey for Panama (ENSPA) 2019. Please use a colon to replace the full-stop before “Results from…”

Abstract: the abbreviation for World Health Organisation and non-communicable diseases should be removed from the Abstract; Organisation (line 12) or Organization (line 36), Please strive to be consistent; please provide a full name for EEE, instead of an abbreviation.

Introduction: please check the sentence in lines 41-43; there are too many paragraphs in the introduction, so please integrate the paragraphs rationally; in lines 82-83, the reviewer suggests using “health and lifestyle factors” to replace “body mass index (BMI), self-perception of health status, life tobacco and alcohol consumption”, and as well as in line 107.

Methods: in lines 87-90, the authors mentioned that the ENSPA comprised blood pressure measurements and blood sampling. The reviewer thinks that blood pressure should be added to this study; additionally, why were other important sociodemographic factors such as incomes and professions not used in this study? they were maybe associated with PI or SB.

Results: please revise the sentences in lines 257-258; there are too many short paragraphs in the Results, so please integrate these paragraphs rationally.

Discussion: there are too many short paragraphs in the Main Findings, so please integrate these paragraphs rationally.

Conclusion: were there no associations between sociodemographic factors, BMI, self-perception of health status, life tobacco and alcohol consumption and PI? The author should mention the above question because this was one of the aims of this study written in lines 81-83.

Round 2

Reviewer 1 Report

First of all I would like to acknowledge and congratulate the authors for their revision work. 

I believe the article will be ready for publication after some minor revisions are suggested.

- Change the title, shorten it. Suggestion: Physical Inactivity and Sedentary Behaviour Among Panamanian Adults: ENSPA 2019 results.

- Change 0.000 to <0.001

Again, congratulations to the authors and I would encourage you to continue this research.

Author Response

Point 1- Change the title, shorten it. Suggestion: Physical Inactivity and Sedentary Behaviour Among Panamanian Adults: ENSPA 2019 results.

Response: Thanks for your suggestion. Although, we think the readers will benefit with a shortened title, we keep the meaning of the acronym ENSPA because this study is new and it is not well known for most of them. Therefore, the title is changed as follows:

Physical Inactivity and Sedentary Behaviour Among Panamanian Adults: Results from the National Health Survey for Panama (ENSPA) 2019.

Point 2- Change 0.000 to <0.001

Response: Thank you for your observation. We changed the “0.000” to “< 0.001” thoroughly in the test and tables of the manuscript.

We thank the Reviewer #1 for raising comments and suggestions that have improved the manuscript quality.

Reviewer 2 Report

Thanks for the revision. The reviewer has no further comments.

Author Response

We thank the Reviewer # 2 for all the comments and suggestions that have improved this manuscript quality.